

# Microbiome succession during ammonification in eelgrass bed sediments

Cassandra L. Ettinger[1], Susan L. Williams[2,3], Jessica M. Abbott[2], John J. Stachowicz[2] and Jonathan A. Eisen[1,2,4]

[1] Genome Center, University of California, Davis, CA, United States of America
[2] Department of Evolution and Ecology, University of California, Davis, CA, United States of America
[3] Bodega Marine Lab, University of California, Davis, Bodega Bay, CA, United States of America
[4] Department of Medical Microbiology and Immunology, University of California, Davis, CA, United States of America

## ABSTRACT

**Background**. Eelgrass (*Zostera marina*) is a marine angiosperm and foundation species that plays an important ecological role in primary production, food web support, and elemental cycling in coastal ecosystems. As with other plants, the microbial communities living in, on, and near eelgrass are thought to be intimately connected to the ecology and biology of eelgrass. Here we characterized the microbial communities in eelgrass sediments throughout an experiment to quantify the rate of ammonification, the first step in early remineralization of organic matter, also known as diagenesis, from plots at a field site in Bodega Bay, CA.

**Methods**. Sediment was collected from 72 plots from a 15 month long field experiment in which eelgrass genotypic richness and relatedness were manipulated. In the laboratory, we placed sediment samples ($n = 4$ per plot) under a $N_2$ atmosphere, incubated them at *in situ* temperatures (15 °C) and sampled them initially and after 4, 7, 13, and 19 days to determine the ammonification rate. Comparative microbiome analysis using high throughput sequencing of 16S rRNA genes was performed on sediment samples taken initially and at seven, 13 and 19 days to characterize changes in the relative abundances of microbial taxa throughout ammonification.

**Results**. Within-sample diversity of the sediment microbial communities across all plots decreased after the initial timepoint using both richness based (observed number of OTUs, Chao1) and richness and evenness based diversity metrics (Shannon, Inverse Simpson). Additionally, microbial community composition changed across the different timepoints. Many of the observed changes in relative abundance of taxonomic groups between timepoints appeared driven by sulfur cycling with observed decreases in predicted sulfur reducers (*Desulfobacterales*) and corresponding increases in predicted sulfide oxidizers (*Thiotrichales*). None of these changes in composition or richness were associated with variation in ammonification rates.

**Discussion**. Our results showed that the microbiome of sediment from different plots followed similar successional patterns, which we infer to be due to changes related to sulfur metabolism. These large changes likely overwhelmed any potential changes in sediment microbiome related to ammonification rate. We found no relationship between eelgrass presence or genetic composition and the microbiome. This was likely due to our sampling of bulk sediments to measure ammonification rates rather than sampling microbes in sediment directly in contact with the plants and suggests that eelgrass influence on the sediment microbiome may be limited in spatial extent. More

Corresponding author
Jonathan A. Eisen,
jaeisen@ucdavis.edu

in-depth functional studies associated with eelgrass microbiome will be required in order to fully understand the implications of these microbial communities in broader host-plant and ecosystem functions (e.g., elemental cycling and eelgrass-microbe interactions).

## INTRODUCTION

Eelgrass (*Zostera marina* L.) is a widely-distributed marine angiosperm that supports ecologically and economically valuable functions (*Williams & Heck, 2001*), including high rates of primary production, higher trophic levels, and elemental cycling (*Hemminga & Duarte, 2000*). Much of the high primary production of eelgrass and its associated epiphytic algal community ends up as detritus (*Cebrian & Lartigue, 2004*), which fuels high rates of ammonification, the first step in early diagenesis (the combination of biological, chemical and physical processes that act on deposited organic matter (*Berner, 1980*)), in the sediments of eelgrass beds. Although the role of microbes in the decomposition of organic matter and remineralization in marine sediments is broadly appreciated (*Arndt et al., 2013*), the extent to which microbial community composition and process rates are influenced by the characteristics of eelgrass beds is unclear. In this study, we looked for correlations between successional patterns in microbial communities during early diagenesis, rates of ammonification in eelgrass bed sediments, and the abundance and genetic composition of eelgrass from plots in Bodega Bay, CA.

The microorganisms associated with eelgrass have been found to be distinct for different eelgrass parts (e.g., roots, leaves, rhizomes) and appear to vary within and between host plants (*Fahimipour et al., 2017*; *Bengtsson et al., 2017*; *Ettinger et al., 2017*; Holland-Moritz et al., 2017, unpublished data). Many of the dominant taxa found in association with eelgrass beds are predicted to be involved in nitrogen and sulfur cycling (*Capone, 1982*; *Welsh, 2000*; *Nielsen et al., 2001*; *Lovell, 2002*; *Sun et al., 2015*; *Cúcio et al., 2016*; *Ettinger et al., 2017*; Holland-Moritz et al., 2017, unpublished data). The remineralization of nitrogen in seagrass sediments is of great importance as it is often the limiting nutrient for seagrass growth; ammonium is the largest pool of dissolved inorganic nitrogen in the sediments and is preferentially taken up by eelgrass (*Dennison, Aller & Alberte, 1987*; *Romero, Lee & Alcoverro, 2006*). The microbial communities in eelgrass bed sediment are significantly different from that of surrounding unvegetated sediment (*Cúcio et al., 2016*) and even from eelgrass roots collected within the same bed (*Fahimipour et al., 2017*; *Ettinger et al., 2017*). Furthermore, even within eelgrass beds, sediment community composition differences are correlated with eelgrass density (*Ettinger et al., 2017*), suggesting the potential for eelgrass influence of microbial processes.

Seagrass density, biomass, growth and resilience are all known to be influenced by the genetic composition and diversity of eelgrass assemblages (*Hughes & Stachowicz, 2004*;

*Reusch et al., 2005*; *Hughes & Stachowicz, 2011*; *Stachowicz et al., 2013*). At the conclusion of a larger experiment testing the effects of eelgrass genotypic richness and relatedness on eelgrass biomass accumulation and other ecosystem functions (*Abbott, 2015*; Abbott et al., 2017, unpublished data), we sampled the microbial communities in sediments in eelgrass plots that varied in genetic diversity. We characterized the relative abundances of microbial taxa and how they changed as early diagenesis proceeded during a laboratory experiment that quantified the rate of ammonification as a function of plant genotypic diversity and abundance.

## METHODS

### Ammonification experiment

The rate of ammonification was determined in sediments collected from plots of a field experiment lasting 15 months in which eelgrass genotypic richness and relatedness were manipulated and various ecosystem functions were measured (*Abbott, 2015*; Abbott et al., 2017, unpublished data). The experiment initially crossed two levels of genotypic richness (2, 6) with three levels of genetic relatedness (more, less, and as closely related as expected by chance (*Frasier, 2008*; *Stachowicz et al., 2013*)) with six replicates per richness × relatedness combination for a total of 72 plots. Plots were 40.4 cm long × 32.7 cm wide × 15.2 cm deep. Genotypic composition changed in the treatments as a result of mortality of some planted genotypes early in the experiment and some plots lost all eelgrass by the end of the experiment; this mortality was independent of treatment. Because samples for ammonification were taken at the end of the experiment, we used final genotypic composition to calculate realized diversity and relatedness for each plot for use in analysis. Eelgrass tissues were collected for the field experiment under California Department of Fish and Wildlife Scientific Collecting Permit # SC 4874.

In October 2014, prior to the harvest of eelgrass from the experiment, we collected ~500 cm$^3$ of sediment from 0–10 cm (the rooting depth) in each plot to determine the rate of ammonification (see Williams et al., 2017, unpublished data for more details). In the laboratory, we placed homogenized sediment samples in a $N_2$- filled glove box, removed macroscopic pieces of eelgrass and animals using forceps, and then filled opaque glass centrifuge tubes with sediments ($n = 4$ per plot). Tubes were incubated at *in situ* temperatures (15 °C) and sampled for porewater and absorbed ammonium and sediment porosity initially and after 4, 7, 13, and 19 days of incubation. Ammonium production rates were calculated by linear regression of $\mu$mol $NH_4$-$N_{porewater+adsorbed}$/L sediment versus incubation time (days) (*Mackin & Aller, 1984*; *Dennison, Aller & Alberte, 1987*; *Williams, 1990*). We also removed belowground and aboveground eelgrass biomass from each plot, cleaned it of sediments and epiphytes, and dried it to constant mass (see *Abbott, 2015*; Abbott et al., 2017, unpublished data for more details).

### Molecular analysis

Sediment was collected at each timepoint during the ammonification experiment for microbial analysis ($n = 72$ per timepoint). DNA was extracted from the sediment taken initially and at 7, 13 and 19 days ($n = 288$; herein referred to as timepoints 1, 2, 3 and 4

respectively) using the PowerSoil DNA Isolation kit (MO BIO Laboratories, Inc., Carlsbad, CA, USA) according to the manufacturer's protocol. The V4 region of the 16S rRNA gene was amplified using the "universal" 515F and 806R primers (*Caporaso et al., 2011*) with a modified barcode system as in *Fahimipour et al. (2017)*. A detailed amplification protocol can be found here (https://figshare.com/articles/Seagrass_Microbiome_16S_ Library_PCR_Protocol_with_PNA_Blockers/5267359). Molecular libraries were sent to the http://dnatech.genomecenter.ucdavis.edu for sequencing on an Illumina MiSeq (Illumina, Inc. San Diego, CA, USA) to generate 250 bp paired-end sequence reads.

## Sequence processing

A custom in-house script was used to demultiplex, quality check and merge paired-end reads (https://github.com/gjospin/scripts/blob/master/Demul_trim_prep.pl). Sequences were then analyzed using the Quantitative Insights Into Microbial Ecology (QIIME) v. 1.9.0 workflow (*Caporaso et al., 2010*). For a detailed walkthrough of the following analysis using QIIME see the IPython notebook (http://nbviewer.jupyter.org/gist/ casett/a42c64ca4b74b1d414f59eb5362e63a3). A total of 10,958,285 reads obtained from the sequencing run passed quality filtering (Q20), of which 7,856,501 paired-end reads merged successfully (71.69%). Chimeras were identified using USEARCH v. 6.1 and filtered out. Sequences were then *de novo* clustered into operational taxonomic units (OTUs) at 97 percent similarity using UCLUST (*Edgar, 2010*) and taxonomy was assigned using the GreenGenes database (v.13_8) (*DeSantis et al., 2006*). Using the filter_taxa_from_otu_table.py and filter_otus_from_otu_table.py QIIME scripts, chloroplast DNA, mitochondrial DNA, singletons and reads classified as "Unassigned" at the domain level were filtered out of the dataset before downstream analysis.

## Data analysis and visualization

Data manipulation, visualization and statistical analyses were performed in R (*R Core Team, 2016*) using the ggplot2 (*Wickham, 2009*), vegan (*Dixon, 2003*), phyloseq (*McMurdie & Holmes, 2013*) coin (*Hothorn et al., 2008*) and FSA packages (*Ogle, 2016*). For statistical comparisons and visualization, the dataset was subsampled without replacement to an even depth of 5,000 sequences. As a result, eight samples were removed from downstream analysis due to low sequence counts (SampleID: I4T4, C5T4, K3T3, J4T3, J2T3, D5T3, G5T4 and K2T3). A depth of 5,000 sequences was chosen to maximize the number of reads per sample while minimizing the number of samples removed from downstream analysis.

A variety of metrics, including observed OTUs, Chao1 (*Chao, 1984*), Shannon (*Shannon & Weaver, 1949*) and Inverse Simpson (*Simpson, 1949*) indices, were used to calculate the within-sample (alpha) diversity for the dataset. Kruskal–Wallis tests with 9,999 permutations were used to test for significant differences in alpha diversity between different sample categories including timepoint (1–4), block (A-L), spot (1–6), plot location (block x spot), eelgrass final richness (0–6)and eelgrass status in the plot (one genotype, multiple genotypes or none present). For categories in which the Kruskal–Wallis test resulted in a rejected null hypothesis ($p < 0.05$), Bonferroni corrected post-hoc Dunn tests were performed.

To assess between-sample (beta) diversity, the Unifrac (weighted and unweighted) (*Lozupone et al., 2007*; *Hamady, Lozupone & Knight, 2010*) and Bray–Curtis (*Bray & Curtis,*

*1957*) dissimilarities were calculated. These diversity metrics were then compared using permutational manovas (PERMANOVAs) to test for significant differences between sample categories (see above) with 9,999 permutations using the Bonferroni correction (*Anderson, 2001*). Mantel tests were used to test for correlations between Bray–Curtis dissimilarities calculated for the microbial data and euclidean distances calculated for continuous variables such as aboveground eelgrass biomass (g/plot), belowground eelgrass biomass (g/plot), total eelgrass biomass (g/plot), plot decomposition rate, detritus standing stock (g/plot), ammonification rate ($\mu$mol $NH_4$-N/L sediment/d) and eelgrass plot final genotypic diversity, trait diversity (e.g., growth, nutrient uptake and photosynthetic parameters), and relatedness (assessed previously in (*Abbott, 2015*; Abbott et al., 2017, unpublished data) with Shannon Diversity, Rao's Q and average relatedness, respectively). These tests were performed in R with vegan using 9,999 permutations.

To compare microbial community composition among timepoints, we collapsed OTUs into taxonomic orders using the tax_glom function in phyloseq and then removed groups with a mean abundance of less than two percent. Rare groups were removed to avoid false positives from low abundance taxa and to focus analysis on abundant groups that may influence sediment biogeochemistry. The average relative abundance of taxonomic orders was compared between timepoints using Bonferroni corrected Kruskal–Wallis tests in R. For taxonomic groups where the Kruskal–Wallis test resulted in a rejected null hypothesis, Bonferroni corrected post-hoc Dunn tests were performed to identify which timepoint comparisons for each taxonomic order were significantly different. To determine the nature of the relationship between ammonification rate and specific taxonomic groups whose mean relative abundances differed significantly between timepoints, we built linear models in R. We focused specifically on the three taxonomic groups with the largest variance in relative abundance and the models were built using the timepoint where the largest variance was observed.

## RESULTS

### Alpha diversity metrics: within sample diversity decreases after initial timepoint

Alpha diversity was significantly different between timepoints (K–W test; $p < 0.001$, Fig. 1, Table S1) for all metrics and post-hoc Dunn tests identified that the alpha diversity for timepoint 1 was consistently greater than that for other timepoints ($p < 0.001$, Table S2). The decrease in diversity from timepoint 1 to the subsequent timepoints is expected as obligate aerobes are not likely to survive after initial inoculation in sealed tubes. Plot location, eelgrass relatedness and eelgrass richness were not correlated with any estimate of alpha-diversity across timepoints or within single timepoints (K–W test, $p > 0.05$, Table S1).

### Beta diversity metrics: microbial community composition changes over time

Microbial community composition differed between timepoints for all three dissimilarity metrics, Bray–Curtis, unweighted and weighted Unifrac (PERMANOVA, $p < 0.001$,

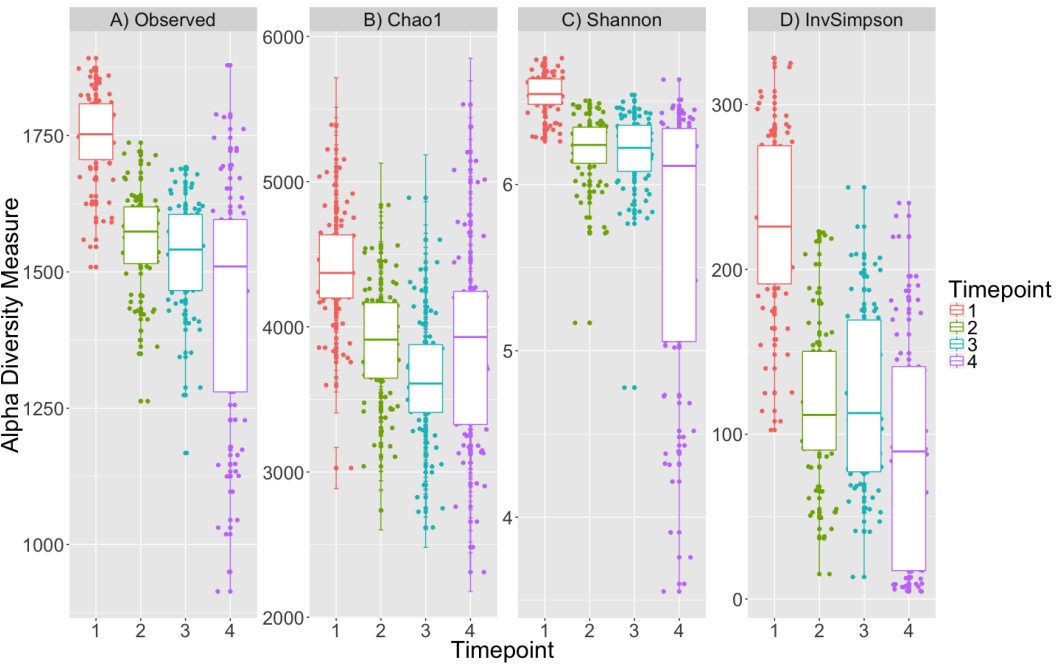

**Figure 1** **Alpha diversity decreases over time.** Four alpha diversity metrics are depicted as boxplots colored by timepoint: (A) observed number of OTUs, (B) Chao1, (C) Shannon and (D) Inverse Simpson indices. Timepoints shown are: 1 (initial samples), 2 (seven days), 3 (13 days), and 4 (19 days).

Fig. 2, Table S3). Subsequent pair-wise PERMANOVA test results found that all pair-wise timepoint comparisons differ significantly in composition ($p < 0.001$, Table S4). PERMANOVA test results for other sample categories were not significantly different ($p > 0.05$, Table S3). Surprisingly, we did not detect any associations of the initial microbiome (timepoint 1) with plot level features such as eelgrass genotypic richness or eelgrass presence/absence (Fig. 3, Table S5).

## Microbial composition effects on ammonification rate

Ammonification rates ranged from 12 to 640 µmol $NH_4$-N/L sediment/d, values typical for eelgrass (*Iizumi, Hattori & McRoy, 1982*; *Dennison, Aller & Alberte, 1987*; Williams et al., 2017, unpublished data). Using the full dataset, we tested for correlations between Bray–Curtis dissimilarities and euclidean distances of several measured variables including ammonification rate and eelgrass final genotypic diversity and relatedness. None of these measured variables were correlated with microbial dissimilarities (Mantel test, $p > 0.05$, Table S6). We then focused our analyses on testing for correlations between these measures and the dissimilarities of only the initial or final timepoints, but still found no correlations (Mantel test, $p > 0.05$, Table S7).

## Taxonomic composition

The orders *Pirellulales*, *Chromatiales*, *Desulfobacterales*, *Bacteroidales*, *Alteromonadales*, *Campylobacterales* and *Thiotrichales* had mean relative abundances that were significantly different over time (K–W test, $p < 0.001$, Table S8). Since we were interested in the

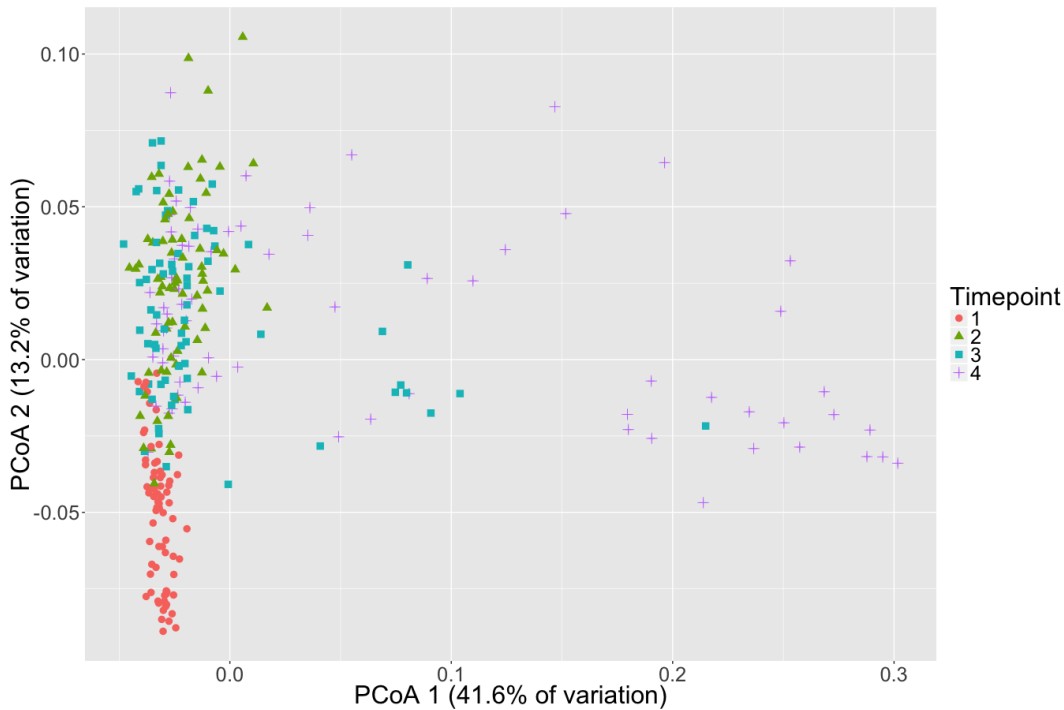

**Figure 2  Microbial community composition changes over time.** Principal Coordinates Analysis (PCoA) of Weighted Unifrac distances of microbial communities are shown here with shapes and colors representative of respective timepoint. Timepoints shown are: 1 (initial samples), 2 (seven days), 3 (13 days), and 4 (19 days).

significance of the directional changes in the observed succession pattern, we focused our investigation on the sequential timepoint comparisons during post-hoc analysis.

We saw a clear succession in eelgrass sediment microbiota during the experiment, which was characterized by several significant differences (Fig. 4, Tables S9, S10). The strongest among these involved several main observations:

1. An initial increase in the mean relative abundance of *Campylobacterales,* mainly members of the family *Helicobacteraceae,* between timepoints 1 and 2 (4.8–12.57%), followed by a decrease in relative abundance (12.57–9.36%) from timepoint 2 to 3.

2. An increase in relative abundance from 3.12 to 6.21% in *Alteromonadales* between timepoints 2–4.

3. A doubling of the average relative abundance of *Thiotrichales,* specifically the genus *Thiomicrospira,* from 9.36 to 18.53% between timepoint 3 and 4.

In our linear model analysis, we did not detect a significant relationship between ammonification rate and the relative abundance of *Thiotrichales* (timepoint 4, $F$-statistic $= 0.323$, adjusted r-squared $= -0.01$, $p = 0.517$), *Alteromonadales* (timepoint 4, $F$-statistic $= 0.167$, adjusted r-squared $= -0.012$, $p = 0.684$) or *Campylobacterales* (timepoint 2, $F$-statistic $= 1.962$, adjusted r-squared $= 0.013$, $p = 0.166$).

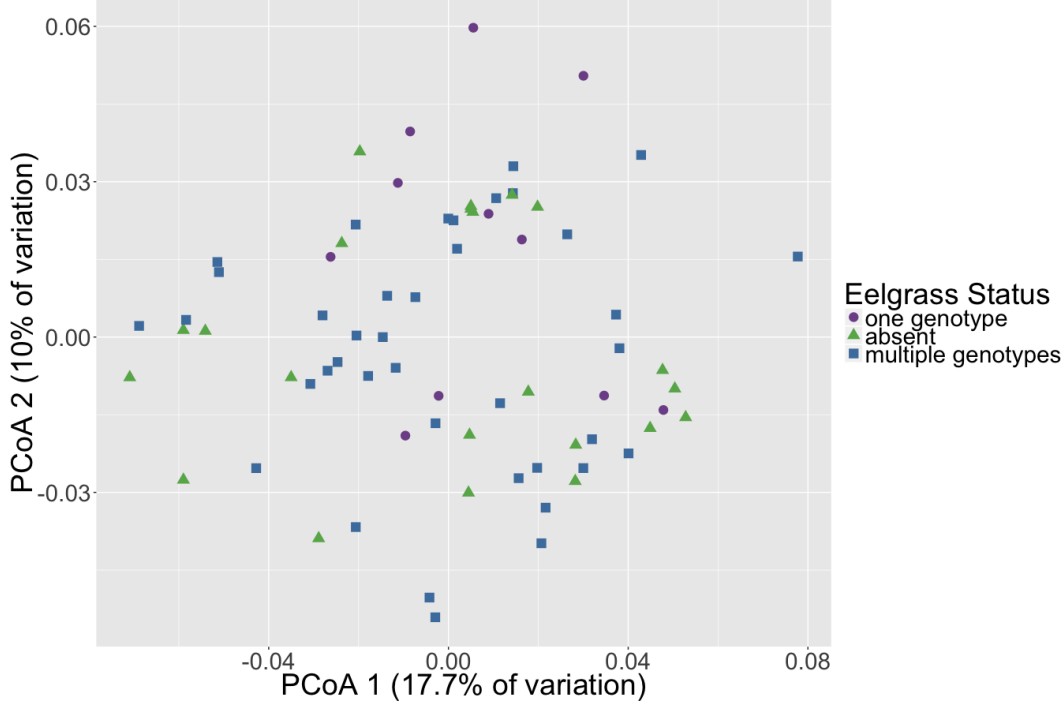

**Figure 3** **Initial microbial community composition is not correlated with eelgrass presence/absence.**
Principal Coordinates Analysis (PCoA) of Weighted Unifrac distances of microbial communities at the
initial timepoint (timepoint 1) are shown here. Points in the ordination are colored by eelgrass status in
each plot (one genotype, multiple genotypes, absent).

## DISCUSSION

We did not detect any association of the microbiome with plot level features such as
eelgrass genotypic richness or eelgrass presence/absence (Fig. 3). This result originally
seemed surprising given previous work indicating a correlation between eelgrass presence
and sediment microbiota (*Cúcio et al., 2016*; *Ettinger et al., 2017*). However, it is important
to note that microbiome samples came from homogenized bulk sediment collected from
whole plots rather than sediment specifically in close association with eelgrass roots.
This suggests that associations between microbiota and eelgrass are localized to plant
surfaces or immediately adjacent sediments do not extend far from the plant itself. Indeed,
*Fahimipour et al. (2017)* found that the root microbiome differed substantially from that
found in sediments taken from within the eelgrass bed, but not specifically associated
with roots. Alternatively, it is possible that the immediate transport from field to the lab
and homogenization fundamentally altered the microbiome, causing the differences with
previous studies. However, the sediments were kept dark and cold in the sediment corers
until extrusion and homogenization under an oxygen-free environment, to preserve the
sediment environment to the degree possible. One further possibility is that these plots do
not differ because eelgrass has a lasting effect on the sediment microbiome and the plots
without eelgrass, since they previously, although briefly, had eelgrass, have just not yet
returned to a non-eelgrass microbiome state.

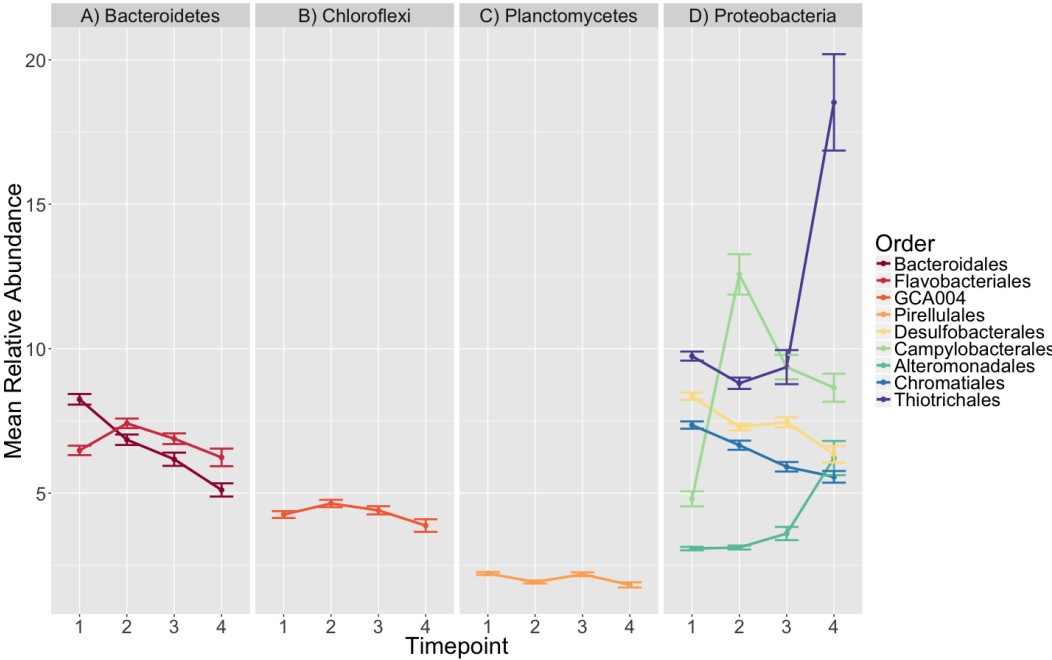

**Figure 4** **Taxonomic composition varies over time.** The average relative abundance of taxonomic orders with a mean greater than two percent are shown across timepoints with the standard error of the mean represented by error bars and lines colored by taxonomic order. Panels group taxonomic orders by phylum (A) Bacteroidetes, (B) Chloroflexi, (C) Planctomycetes and (D) Proteobacteria. Timepoints shown are: 1 (initial samples), 2 (seven days), 3 (13 days), and 4 (19 days).

To conduct the ammonification experiment, the sediment was moved from its natural setting, in which a micro-oxic zone exists around eelgrass roots (*Jensen et al., 2005*), into an anaerobic, enclosed system. Seagrass sediments are highly anaerobic below the very top layers and thus, organic matter diagenesis is predominantly an anaerobic process (*Harrison, 1989*; *Marbà et al., 2006*). This procedure enabled us not only to quantify ammonification rates but also to study successional shifts in communities under these conditions during which we observed reductions in alpha diversity and changes in taxonomic composition. We note that each "replicate" sample does not follow the exact sample succession pattern. This can be seen especially in timepoint 4 samples which are widely scattered on the PCoA plot (Fig. 2). The variation between these samples appears to be due in large part to differences in relative abundance of specific likely sulfide oxidizers (e.g., *Thiotrichales*). We also note that by conducting this process in an anaerobic setting and only focusing on 16S rRNA gene sequence analysis, we are unable to detect the role of microbial eukaryotes (e.g., fungi, ciliates, amoeba) during and throughout early diagenesis in seagrass bed sediments. This may be of little consequence as, in contrast to in terrestrial systems where microbial eukaryotes are known to participate in ammonification, these groups are historically thought to contribute little to the primarily anaerobic process of organic matter diagenesis in seagrass sediments (*Newell, 1981*) for *Z. marina*; (*Blum et al., 1988*) for tropical seagrass leaf litter, (*Harrison, 1989*). However, it is important to note that marine

microbial eukaryotes have been observed in seagrass detritus (*Harrison & Mann, 1975*; *Harrison, 1989*) and very little is known about the functions of these microorganisms.

The different samples, regardless of the ammonification rate, followed similar successional patterns, which we infer to be due largely to a response to sulfur metabolism, based on the predicted functional roles of the taxonomic groups that exhibited the greatest change in relative abundance across timepoints. For example, the relative decrease in *Desulfobacterales* concomitant with an increase in *Thiotrichales* and *Alteromonadales* we believe is likely the result of the coupling of sulfate reduction and sulfide oxidation during the experiment. We have inferred this based on examination of the literature regarding these groups. Most of the characterized species in the *Desulfobacterales* are sulfate-reducing bacteria that commonly reduce sulfate to sulfide. Species in this group have been isolated from a variety of marine and freshwater habitats (*Kuever, Rainey & Widdel, 2015*). The *Thiotrichales* are broadly known as filamentous sulfur-oxidizers (*Garrity, Bell & Lilburn, 2005*), are the dominant sulfur-oxidizers in salt marsh sediments (Thomas et al., 2014) and are enriched in eelgrass-associated sediment (*Ettinger et al., 2017*). *Thiomicrospira,* specifically, are mesophilic sulfur oxidizers that oxidize thiosulfate to sulfate and elemental sulfur in marine ecosystems (*Kuenen, Robertson & Tuovinen, 1992*; *Scott et al., 2006*; *Sievert et al., 2008*). Although relatively little is known about the ecology of *Alteromonadales,* some isolates have been found to reduce sulfate and thiosulfate to sulfide (*Semple, Westlake & Krouse, 1987*; *Bowman & McMeekin, 2005*). Members of this group can also degrade dimethylsulfopropionate, an osmoprotectant produced by marine algae, phytoplankton and some vascular plants including seagrasses, to dimethyl sulfide (*Jonkers, Van Bergeijk & Van Gemerden, 2000*; *Ansede, Friedman & Yoch, 2001*). Additionally, *Thiotrichales* and *Alteromonadales* species have been suggested to work in concert to degrade marine dissolved organic matter in seawater (*McCarren et al., 2010*). It is worth noting, however, that members of *Alteromonadales* have been previously observed to dominate during succession in mesocosms (*Schäfer, Servais & Muyzer, 2000*).

We did not detect any major correlations between the microbiome and ammonification rate. There are multiple potential explanations for this including that ammonium production can occur as a byproduct of a variety of microbial processes and metabolic pathways (*Herbert, 1999*; *Zehr & Kudela, 2011*). General microbial activity has been previously linked with rates of seagrass decomposition (*Blum & Mills, 1991*), so perhaps what we observe here is a broader community process that cannot be linked to any one taxonomic group. A more likely explanation is that the effects of ammonium production may be present in our dataset, but are masked here by stronger processes (e.g., sulfur metabolism) that are independent of eelgrass characteristics. In marine sediments, sulfate reduction can be attributed as responsible for a large part of organic carbon oxidation and the dominant anaerobic process as it is more thermodynamically favorable than methanogenesis (*Berner, 1980*; *Capone & Kiene, 1988*; *Marbà et al., 2006*). Additionally, fermentative microorganisms break down marine organic matter into lower molecular weight organics in concert with the sulfate reducers that use the products (*Berner, 1980*). Thus, the overall succession pattern that we are seeing is likely an accurate representation of

what occurs during early remineralization of organic matter in anoxic seagrass sediments even if we cannot link it to the ammonification rate here.

## CONCLUSIONS

Seagrass beds are hotspots of primary production, organic matter degradation, and elemental cycling and previous work has suggested that sulfur metabolism can play an important ecological role in these beds. In this study, we wanted to identify if successional patterns in microbial communities during early diagenesis were correlated with the rate of ammonification. We found no such correlation, instead, observing a successional pattern more consistent with sulfur cycling as the dominant biogeochemical process. Future work should endeavor to use metagenomic techniques to investigate the abundance of genes associated with sulfur metabolism to confirm this observation. Additionally, although no correlation was found between ammonification rate and 16S rRNA gene sequence data, metagenomics or metatranscriptomics might identify functional genes that are enriched in samples with a higher rate of ammonification. Seagrass beds have important ecosystem functions, but our knowledge of the microbial communities inhabiting these beds and their functions is still fragmentary. This work contributes to the growing body of knowledge on the eelgrass microbiome, providing some contextual functional framework for the sediment associated generally within these beds and highlighting a growing need for functional studies in this and other host-microbe-environment systems.

## ACKNOWLEDGEMENTS

Illumina sequencing was performed at the DNA Technologies Core facility in the UC Davis Genome Center in Davis, California. We thank Qingyi ''John'' Zhang for his help with the DNA extractions and Illumina library preparation.

### Funding

This work was funded by a grant from the Gordon and Betty Moore Foundation (GBMF333) ''Investigating the co-evolutionary relationships between seagrasses and their microbial symbionts.'' The funders had no role in study design, data collection and analysis, decision to publish, or preparation of the manuscript.

### Grant Disclosures

The following grant information was disclosed by the authors:
Gordon and Betty Moore Foundation: GBMF333.

### Competing Interests

Jonathan A. Eisen is an Academic Editor for PeerJ.

## Author Contributions

- Cassandra L. Ettinger analyzed the data, wrote the paper, prepared figures and/or tables, reviewed drafts of the paper.
- Susan L. Williams conceived, designed and performed the ammonification experiment, reviewed drafts of the paper, helped write the paper.
- Jessica M. Abbott conceived, designed and performed the eelgrass field experiment, reviewed drafts of paper, helped with ammonification experiment.
- John J. Stachowicz reviewed drafts of the paper, advised on experimental design, edited drafts of paper.
- Jonathan A. Eisen contributed reagents/materials/analysis tools, reviewed drafts of the paper, advised on data analysis, edited drafts of paper.

## Field Study Permissions

The following information was supplied relating to field study approvals (i.e., approving body and any reference numbers):

Eelgrass tissues were collected for the field experiment under California Department of Fish and Wildlife Scientific Collecting Permit #SC 4874.

## DNA Deposition

The following information was supplied regarding the deposition of DNA sequences:

This 16S rRNA sequencing project has been deposited in GenBank under accession no. PRJNA350672.

## Data Availability

Lang, Jenna (2015): Seagrass Microbiome. figshare.
https://doi.org/10.6084/m9.figshare.1598220.v1.

## Supplemental Information

Supplemental information for this article can be found online at http://dx.doi.org/10.7717/peerj.3674#supplemental-information.

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
