# Peer review of "Microbiome succession during ammonification in eelgrass bed sediments"

_PeerJ, doi:10.7717/peerj.3674_

## Round 0.1 · original submission · Major Revisions

Thank you for submitting your manuscript to PeerJ. After careful consideration, we feel that it has merit but requires major revisions as it currently stands. Therefore, we invite you to submit a revised version of the manuscript that addresses all the points raised by the reviewers, specially by the reviewer 2.

Reviewer 1 ·

Basic reporting

The paper is informative and interesting, and I am very supportive of the general approach of trying to measure and link characteristics of the plant host, environment and the microbiome. The study was well developed and the sampling design and methodology appropriate for the objectives of the study.

The ms is clear and well written, professional English used throughout.

Good use of figs and tables.

Experimental design

(4) Line 175- Please explain, what are the criteria for selecting the subsampling depth of 5000 sequences?

Validity of the findings

I think the statistical approach was well carried out. However, the raw data of the ammonification rates and the whole microbiome composition should be presented to better understand the variability of these datasets.

Additional comments

This ms compares changes in eelgrass sediments associated microbiome overtime in response to ammonification rates and eelgrass genotypic richness and relatedness. The paper is informative and interesting, and I am very supportive of the general approach of trying to measure and link characteristics of the plant host, environment and the microbiome. The study was well developed and the sampling design and methodology appropriate for the objectives of the study. Having said that I think a better job can be done with the presentation and interpretation of the data. I highlight several general comments below, which could be addressed in a revision:

(1) Lines 95 -97- The authors’ basic interest was to measure the effects of ammonification in the eelgrass dominated sediment associated microbiome given that nitrogen and sulphur cycling microbial taxa has been reported to be found in eelgrass sediments. I think the authors need to expand about the selected approach, particularly the selection of ammonification activity.

(2) Line 144- Please review and correct, the first day of sampling (day 4) is missing, please include it.

(3) I think that it will be very informative to present in supplementary materials the detailed results of the ammonification rates as well as the whole microbial community composition found associated with the seagrass sediments.

(4) Line 175- Please explain, what are the criteria for selecting the subsampling depth of 5000 sequences?
After these points are addressed the manuscript can be accepted for publication in PeerJ journal.

Reviewer 2 ·

Basic reporting

The research paper “Microbiome succession during ammonification in eelgrass bed sediments” by Ettinger et al, investigates the possible correlation between ammonification rate and microbial communities in eelgrass sediments.

Experimental design

The experimental design, utilizing 16S surveys, for initial biological questioning is acceptable, although correlations between taxonomic information from 16S rDNA and biological processes (.i.e. ammonification) are extremely limitedThe authors, at the end of the paper, proposed a more in depth investigation using metagenomics (in this case, I strongly suggest incorporation of metatranscriptomics), they certainly could have probed the 18S, ITS and Archean 16S to enhance our understanding of this ecological system.

Validity of the findings

Although the authors did not find correlation between the ammonification rate and the microbial communities in eelgrass sediments, probably due to the experimental setup, they were able to suggest sulfur cycling as a factor driven the microbial succession during the experiment. The data and its analysis are strong and sound which provided a reasonable first step towards a more comprehensive understanding for the system.

Additional comments

1) In order to make the study aims clearer, I suggest to transfer the statement in lines 329-330 “In this study, we wanted to identify if successional patterns in microbial communities during early diagenesis were correlated with the rate of ammonification” to the introduction section.

2) Line 148: It would be better to cite the original paper that first described the primers. Caporaso JG, Lauber CL, Walters WA, Berg-Lyons D, Lozupone CA, Turnbaugh PJ, Fierer N, Knight R (2010c) Global patterns of 16S rRNA diversity at a depth of millions of sequences per sample. Proc Natl Acad Sci U S A 108:4516–4522. doi:10.1073/pnas.1000080107

3) Lines 175: please explain why did you choose 5000 reads per sample as a basis for rarefaction.

4) Delete “Overal” from line 64 and 291.

5) Because the sampling influenced the results, as stated in lines 272-274, please provide more detail on how many samples were used for molecular analysis and if the four replicates collected from each plot were homogenized before incubation. If so, why homogenize the samples? Wouldn’t it be better to preserve the stratification of the sediment? Justify.

6) Please provide the accession number to the data set.

7) Line 306. The authors must be careful with a statement like this, since the eukaryotes could have a more important impact to the process in aquatic systems than they do in terrestrial systems. Currently we simply don’t have enough data to back up this statement.

8) Line 318. …”effects of ammonium production may be present in our dataset…” would a higher coverage sequencing depth answer this question?


9) An in depth discussion about the available genomes of Desulfobacterales, Alteromonadales and Thiotrichales would greatly improve the reach of the manuscript.

10) The results are limited to 16S survey with limited biological discussion power at this stage. To answer the specific biological question proposed by the authors a better approach is definitely needed (metagenomics, as stated by the authors, but also metatranscriptomics).

11) If at least, the authors provided a broader approach and analysis (18S, ITS, Archeal 16S, etc) it would add a lot more to broaden the discussion. This would help not only to access the ammonification process (the original question) but also to understand the sulfur cycle in this system (the new-found observation). Just the taxonomic description says very little about the processes in any given system.


12) I could not find the figure legends!

13) The supplemental tables on the supplemental material need footnotes and legends to be interpreted without further assistance.

---

## Round 0.2 · Minor Revisions

The rebuttal letter the additions to the text improved the manuscript significantly.

I agree with reviewer 2 that "the experimental design, utilizing 16S surveys, for initial biological questioning is acceptable. However, correlations between taxonomic information from 16S rDNA alone and biological processes are limited the authors build a strong case for bacterial succession in this system".

We understand the authors´ position that this work is a first step towards a better understanding of the ammonification process in this particular system. And this statement should be very clear in the text (especially in the conclusion). The correlation found between the sulfur cycling and the bacterial community succession is of interest and therefore this MS merits publication. But it still requires Minor Revisions.

Reviewer 2 ·

Basic reporting

The research paper “Microbiome succession during ammonification in eelgrass bed sediments” by Ettinger et al, investigates the possible correlation between ammonification rate and microbial communities in eelgrass sediments.

Experimental design

The authors reply clarify the points of concern raised by the first assessment of the paper. The experimental design, utilizing 16S surveys, for initial biological questioning is certainly acceptable. Although correlations between taxonomic information from 16S rDNA alone and biological processes (i.e. ammonification, sulfur cycle, etc) are limited the authors build a strong case for bacterial succession in this system and a possible correlation with the sulfur cycle.

Validity of the findings

My initial evaluation of this manuscript still stands (below). Nevertheless, the modifications to the text on the R1 version, especially to the discussion section substantially improved the manuscript. Although the authors did not find correlation between the ammonification rate and the microbial communities in eelgrass sediments, probably due to the experimental setup, they were able to suggest sulfur cycling as a factor driven by the microbial succession during the experiment. As stated before, the data and its analysis are strong and sound which provided a reasonable first step towards a more comprehensive understanding of the system.

Additional comments

Line 50: Include space between “n” and “=”.
Line 115: Replace “review” by “revision”.
Line 128: Parentheses are duplicated.
Line 170: add space “denovo”.
Line 274: Include space between “=” and “0.517”.
Line 339: An “o” is missing in “Altermonadales”.

---

## Round 0.3 · accepted · Accept

The paper improved a lot, and it is of interest. Therefore this paper merits publication.

Congratulations!